# A qualitative study of the first batch of medical assistance team's first-hand experience in supporting the nursing homes in Wuhan against COVID-19

**Xiu-Fen Yang[☯], Meng-qi Li[☯], Lu-lu Liao[☯], Hui Feng[iD][‡]\*, Si Zhao[‡], Shuang Wu[‡], Ping Yin[‡]**

Xiangya Nursing School of Central South University, Changsha City, Hunan Province, China

☯ These authors contributed equally to this work.
‡ These authors also contributed equally to this work.
\* 1757083108@qq.com

## Abstract

### Background

The pandemic of coronavirus disease 2019 (COVID-19) has global impact, Wuhan in Hubei province is a high-risk area. And the older people in nursing homes are the most susceptible group to COVID-19. The aim of this study was to describe the practice and experience of the first-line medical team, to provide insights of coping with COVID-19 in China.

### Method

This qualitative study used a descriptive phenomenological design to describe the experience of medical staff supported the nursing homes in Wuhan fighting against COVID-19. Unstructured interviews via online video were conducted with seven medical staffs who supported the nursing homes in Wuhan. Data were analyzed using content analysis in five main themes: for nursing homes, we interviewed the difficulties faced at the most difficult time, services for the older people, and prevention and management strategies, for the medical staff, the psychological experience were interviewed, and the implications for public health emergencies were also reported.

### Conclusions

It is imperative that effective preventive and response measures be implemented to face the outbreak of COVID-19 and meet the care needs of older people in the context of COVID-19.

### Implications

Findings will inform managers of some reasonable instructional strategies for implementing effective infection management. Nursing homes need to provide targeted services to help alleviating their bad psychology for residents.

**Data Availability Statement:** All relevant data are within the manuscript and its Supporting Information files.

**Funding:** Unfunded studies. The author(s) received no specific funding for this work.

**Competing interests:** The authors have declared that no competing interests exist.

## Introduction

The coronavirus disease (COVID-19) has led to the diagnosis of more than 100 million people in more than 200 countries [1]. As COVID-19 a new virus with limited information regarding risk factors, state of emergencies has been declared in many countries around the world. Based on available information and clinical expertise currently, older people aged 65 and over, and those who live in long-term care facilities might be at higher risk for COVID-19. The number of confirmed cases and death toll from the disease kept climbing as the coronavirus pandemic continues to rage across the world. The elderly are undoubtedly the most vulnerable ones, with a higher chance of developing severe or critical conditions, even dying. However, more than 70 percent of patients at 80 years or older from Wuhan recovered, which brings hope to the treatment of older patients worldwide [2].

A painful fact that cannot be ignored is that once the older people are infected, it will be life-threatening. The decline of immunity, the impact of basic chronic diseases, psychological and cognitive disorders, all made the older people susceptible to this virus, with lower cure rate and higher mortality [1]. As a result of working in a high-risk environment, many of the staff in nursing homes tested positive for nucleic acid, and hundreds more were suspected to be under quarantine observation [3], it is difficult for officials to count the actual death toll, the lives of medical staff and the older people are seriously threatened [4].

At the time of the epidemic, there were more than 20, 000 elderly people living in nursing homes in Wuhan with only less than 3,000 medical staff. In addition, with some of them were infected or quarantined, medical staff were in a shortage condition there. Under such circumstances, 168 medical workers from three provinces across China were dispatched to aid the nursing homes in Wuhan in fighting the epidemic after the coronavirus outbroke. The assistance effectively relieved the medical resource shortage in the region and enabled local nursing homes to offer targeted care to the older people with various symptoms, which increased people's chances of recovery.

As the nursing homes are facing the same challenge globally [5], the work experience of these front line medical staff is of great significance. The purpose of the current study included the following: (1) to establish an understanding of the practical first-hand experience of first batch of medical staff in nursing homes; (2) to describe the current difficulties faced by the nursing homes; (3) to provide suggestions and experience as references for the front line medical staff in the nursing homes in the context of epidemic environment.

## Materials and methods

### Aim

We aimed to describe the practice and experience of first-line medical team, at the mean time, to provide insights to COVID-19.

### Participants

The researchers in this study were mainly master candidates and the supervisor was a professor with PhD degree in a medical college in China, all researchers were women. Researchers got a name list of the assistance team to Wuhan, Hubei province from the Chinese government network, during January 22[nd] to March 1[st], 2020 (http://www.gov.cn/). Purposive sampling was used to select staffs from first line medical assistance team with different occupations, who supported the nursing homes in Wuhan against COVID-19 at the time of the most serious epidemic and lack of manpower (January 22[nd] to March 1[st], 2020), and to ensure a broad perspective of medical members of different positions. The first-line medical assistance team consisted

of 26 members, including clinical nurses, doctors, nursing managers and nursing assistants. The medical assistance team were mainly from the department of infection, respiratory and critical care medicine, cardiovascular department, neurology department and other departments, forming a general practice team, which undertake the treatment and management of a nursing home, reducing contact with the outside. They all have many years of clinical experience, before they enter the nursing home, they will be specially trained by the medical staff of the infection department.

The researchers contacted these first-line medical members by phone or text message, the participants who were willing to participate in the study replied their interest to the researchers, then the researchers negotiated time with participants for interview. We didn't stop to contact the front-line support specialists for interviews until the novel themes saturated [6, 7].

## Procedures

At the beginning of the study, a research group was formed under the guidance of the professor. All the members of the group were sophomores majoring in nursing and had completed the master's course, and four of them have participated in qualitative research. Next, the research team collected the preliminary data from the following aspects: The related rate of COVID-19 (incidence rate, mortality rate, cure rate in the whole China/Wuhan City), the predicament faced by the pension institutions, the social support available for the elderly institutions, and the assistance to the medical group. After completing the above preparatory work, we conducted a pilot interview in March 2020 to test the interview questions, prompts, guide [8], and form with a front-line support specialist. Then the researchers adjusted the interview outline according to pilot study, and the detail interview outline attached in the Table 1.

Semi-structured interviews were carried out by two bachelor of nursing researchers who have done qualitative research in the past. One to one WeChat video interviews were used for all participants between March 15–22, 2020. The qualitative data source is mainly from video interviews. Before conducting the interview, we explained the research purpose and data usage to the interviewee, and the oral consent were all obtained. The interview would be conducted later, otherwise the interview would not continue. With the consent of the participants, the participants were asked a series of questions about their demographics, challenge, and experience with fighting against COVID-19. The interviews were held in private rooms where were only interviewers and interviewees, and the time ranged from 30 to 40 min. All interviews were audio recorded and later transcribed verbatim by one investigator (YXF). Besides, the

**Table 1. Interview questions for the first-line medical support specialists.**

| Interview questions |
|---|
| • Could you please talk about what you have done in supporting the nursing homes in Wuhan to fight the COVID-2019? |
| • In your process of understanding and on-the-spot guidance to the nursing homes in Wuhan, what do you think are the difficulties encountered by the nursing homes? How were resolved? |
| • Do you feel pressured during the process of guiding the fight against the COVID in the nursing homes in Wuhan? If so, what are your pressures? How did you solve them? |
| • What are the most impressive things in the process of supporting the nursing homes in Wuhan? Could you share with examples? |
| • In this anti-epidemic process, what kind of services do you think we need to provide for the elderly? Can you give an example? |
| • What do you think are the aspects of the nursing homes in Wuhan that have done a good job in the fight against the epidemic? What needs to be improved? |
| • How does this incident inspire the nursing homes to respond to such public health emergencies in the future? |

first author truthfully documented the responses, expression and gesture of every participants during interviews.

## Ethic statement

Because the project was interviewed in the most serious time of COVID-19. The whole Chinese people were isolated at home and were not allowed to gather. So we adopted video interviews, and did not directly contact with the interviewees. However, before conducting the interview, we explained the research purpose and data usage to the interviewee, and the oral consent were all obtained. Ethical Approval was gained from the Nursing and Behavioral Medicine Institutional Review Boards, Xiangya School of Nursing, Central South University (Approval number: E202037).

## Data analysis

Data analysis and interpretation were guided by Elo and Helvi phases of content analysis [9]. Content analysis includes many specific methods to guide the analysis of data. Based on our research content, purpose, and the guideline, we chose the inductive approach, which can be applied to situations where previous knowledge of this phenomenon is insufficient or fragmented. The analysis then followed seven stages of organizing, immersion, and coding the data, generating categories and themes, offering interpretation through analytic memos, searching for alternative understandings, and writing the report [10, 11]. The researchers used Nvivo version 11 to analysis the transcribed interviews and there were revised during the repeat process. This dilemma and any other disputes in the coding were resolved through discussion and agreement. The researchers identified important data from a data corpus. The data came from different forms: mainly include interview record and posture, sometimes it may include some other information introduced or provided by the interviewee, such as: books, pictures, photographs, statues, ideas, papers, behaviors, etc. The aim of the researcher is to analyze the content of each data item. In most content analyses, we use coding systems to identify and categorize various data items.

The trustworthiness of the study was established by the following criteria: credibility, transferability, dependability, confirmability, and authenticity [12]. Credibility in the study was achieved by taping the interviews [13]. Four researchers and specialists in aged care formed an auditing panel in order to discuss the findings. The transferability of the study was improved by citing excerpts from the interviews to support the findings, looking for evidence to validate the findings, and delving deeper into the social context of Chinese nursing homes. To ensure the dependability of the analysis in this study, the transcripts were analysed by two researchers, and the transcripts and thematic interpretations were also returned to the participants for verification [11]. The researchers established the confirmability of the study by linking explanations with the participants' quotes. Finally, the verbatim transcripts were used to prove the authenticity of the study.

When formulated categories by inductive content analysis, the researcher came to a decision, through interpretation, as to which things to put in the same category. Each category was named using content-characteristic words. Subcategories with similar events and incidents were grouped together as categories and categories were grouped as main categories [14–16].

## Results

### Participants

We conducted online video interviews with seven medical staff supporting the nursing home in Wuhan from March 25 to 31, 2020. Characteristics of the seven participants were shown in

**Table 2. General information of research objects.**

| Code | Age | Sex | Education level | Field of work | Professional level | Professional years | Support Wuhan time | Support time in Wuhan |
|------|-----|-----|----------------|---------------|--------------------|--------------------|--------------------|----------------------|
| **P1** | 48 | F | Bachelor degree | Administrator (Public health management) | Intermediate title | 29years | Feb 22 (2020) | 30 days |
| **P2** | 60 | M | Doctor degree | Administrator (Westernmedicine physicians) | Senior title | 37years | Feb 20 (2020) | 33 days |
| **P3** | 51 | F | Bachelor degree | Administrator (Public health management) | Technical master | 29years | Feb 27 (2020) | 27 days |
| **P4** | 47 | M | Bachelor degree | Administrator (Western medicine physicians) | Intermediate title | 23years | Feb 23 (2020) | 31 days |
| **P5** | 29 | M | Bachelor degree | Doctor of traditional Chinese Medicine | Junior title | 5 years | Feb 23 (2020) | 33 days |
| **P6** | 31 | F | Bachelor degree | Nurse | Senior title | 10 years | Feb 22 (2020) | 30 days |
| **P7** | 26 | F | Bachelor degree | Nurse | Junior title | 6 years | Feb 22 (2020) | 30 days |

F, Female; M, Male.

Note: The coding is in the order of interview and has nothing to do with other information.

Table 2. Among them, two were professional fields of public health, two were managers of western medicine physicians, two were front-line clinical nurses and one was clinical doctor of traditional Chinese medicine. The average age is 47 years (26–60). There were three males (42%), four females (58%). The average working days were 30 days, and the average working years were 20 years.

Based on our literature review and interviews, we extracted five generic categories and 19 sub categories about the experience of fighting COVID -19 in nursing homes (Table 3), as follows:

**Table 3. The category of content analysis.**

| Main theme | Sub-theme |
|------------|-----------|
| Difficulties Faced by the Nursing Homes | Nursing assistants lack of knowledge |
| | Lack of equipment |
| | Lack of experience |
| | Shortage of staff |
| Psychological Experience | Pressure from many sources |
| | Team-cooperation |
| | Challenge |
| Service for the Older People | Providing epidemic prevention knowledge |
| | Traditional Chinese medicine health care |
| | Mental consolation |
| | Basic daily care |
| Prevention and Management Strategies | Establish contingency plans |
| | Health assessment |
| | Targeted training |
| | Single isolation |
| Strategies for Public Health Emergencies | Physical space design |
| | Routine training |
| | Routine training |

## Main theme 1: Difficulties faced by the nursing homes

**Sub-theme 1: Nursing assistants lack of knowledge.** The staff of the organization had weak awareness of epidemic prevention, and the work of epidemic prevention is not standardized:

"*When we arrived at the support organization, we found that the organization did not carry out strict in and out management, and there were three career who could go home, because there were elderly people in need of care*" (participant 1).

The protection of the organization's staff is unqualified, the wearing and taking off protective equipment such as isolation clothing, masks and goggles, do not meet the professional standards:

"*The organization should strictly implement the three-level prevention and control of infectious diseases, which means that the clothing is double-layer, that meaning, isolation clothing, protective clothing, masks and goggles, gloves and shoe covers are two pairs. A suit of protective clothing could be used for three or four days, and some people washed the protective clothing with water*" (Participant 1).

**Sub-theme 2: Lack of equipment.** Isolation clothing, masks, goggles and other isolation equipment specifications did not meet professional protection standards in a limited number:

"*At their worst, the paramedics were also infected and moved away, there was a severe shortage of staff and supplies, and they might just have a mask that they didn't even wear gloves, and that's really how they came into contact with the infected elderly*" (participant 4).

These nursing homes all lacked three areas, two channels and isolation observation rooms:

"*There were no three areas, two channels and isolation observation room in the nursing home, once confirmed, they would be sent to the designated hospital for treatment*" (participant 5).

**Sub-theme 3: Lack of experience.** The outbreak of COVID-19 is sudden. Everyone is slowly exploring its prevention and control, and there is no authoritative and feasible standard:

"*The prevention and control of the epidemic in the elderly care institutions had never been touched before, so we had no experience. All of us were going in teams on a temporary basis, there was not enough communication and tacit understanding between teams*" (participant 6).

**Sub-theme 4: Shortage of staff.** Most of the care staff in the nursing homes, affected by their own cultural level and professional fields, can't provide timely and accurate standard care for the elderly and self-protection during the epidemic:

"*In the existing institutions, there were a lot of nursing assistants, who may not have many educational opportunities at the cultural level. In the process of implementing management standard of hospital infection control, they had not strong cognition of hospital sense, therefore, the implementation did not conform to the specifications*" (participant3).

## Main theme 2: Psychological experience

**Sub-theme 1: Pressure from many sources.**    During the anti-epidemic period, the main pressure mainly includes two groups, the elderly and the staff of the institution. Managers use the Chinese version of self rating anxiety scale [17] to evaluate the psychological status of the elderly and staff, and found that most of them suffer from varying degrees of anxiety.

A *manager said that*: *"During the period of isolation, we randomly selected 90 elderly people with clear cognition from the institution for evaluation, and found that 27 of them were at severe anxiety, 39 were at moderate anxiety, and the remaining 24 were at mild anxiety. Among the 47 staff members, 35 person were at moderate anxiety, 8 person were at mild anxiety, and only 4 person no anxiety"* (Participant 1).

The older people were worried about being infected because someone nearby was diagnosed and sent away.

"*There were some old people who could not fully understand our prevention and control requirements, theory they all understood, but once this measure was implemented on them, thy would be impatient, anxious, swear and cry, so some old people could only limited cooperation, some old people completely did not cooperate*" (Participant 3).

The pressure of staff mainly comes from the support institutions and their daily work. Long time high intensity and high load work, physical overdraft, tiredness and psychological burden:

"*In fact, our pressure was mainly two aspects, the first was the prevention and control of the epidemic situation in pension institutions, we had not been exposed before, so there was no great assurance. The second was that the epidemic prevention work was not like charging on the battlefield, in the battlefield you go forward, not afraid of death are no problem. If one of our players infected, it means all of us members of the team couldn't do any work and to be sequestered*" (Participant 1).

**Sub-theme 2: Team-cooperation.**    The support staff mainly includes four categories: doctors, nurses, nursing assistants for the aged and civil servants of the government.

"*After we arrive in Wuhan, if some policies or processes are difficult to be widely used, we need to use the superior departments. When the work is promoted, if there is any problem or contradiction, the government department will directly order and urge the following implementation*" (Participant 6).

It is also important to share experiences among different support teams.

"*because we had achieved remarkable results in the support process, the government asked us to share their experience with Wuhan Qiaokou District Welfare Institute and participated in the formulation of some rules and procedures of the Institute*" (Participant 5).

**Sub-theme 3: Challenge.**    As medical professionals, combined with own field skills, scientific protest, in the process of anti-epidemic advice, give correct and effective guidance:

"*After we arrived at the nursing home, we found that the institution epidemic prevention and disinfection work was unqualified, I used to work in the department as a head of infection*

*management, responsible for disinfection and isolation, familiar with infection management work, assessment of institution epidemic prevention work not in place, to help the institutions leaders" (participant 3).*

## Main theme 3: Service for the older people

**Sub-theme 1: Providing epidemic prevention knowledge.**   Most of the elderly in the institution are blind to related preventive knowledge, we combined the characteristics of the elderly to explain the knowledge of infectious diseases and protective measures:

"*After the training, the elderly improved their consciousness, and then the disinfection and iso-lation were standard. Our colleagues would pass some better news information about COVID-19 to their old people every day when they went to work, and then they would teach the elderly learn some knowledge about the prevention of virus and guide them to do well in self-protection" (Participant 7).*

**Sub-theme 2: Traditional Chinese medicine health care.**   In the process of fighting the epidemic, traditional Chinese medicine has played a certain role in the treatment and prevention:

"*During the epidemic, it was forbidden for the elderly to go out for activities. We would teach the elderly health care points, let them massage themselves, teach them moxibustion methods, or help them with moxibustion when everyone's protection was qualified. Then there was the health preserving skill of traditional Chinese medicine, Baduanjin, which could help the elderly adjust their physical and mental state".*

**Sub-theme 3: Mental consolation.**   To solve pressures faced by the elderly, it is important to take effective measures:

"*Considering that these elders had been isolated for more than 40 days, we also used many ways to give them some psychological comfort. . .. . .so we had created some conditions in this area. In addition, we download some graffiti from the Internet–Secret Garden, then guided the elderly to paint in their own room, and then made some handmade origami paintings" (Participant1).*

**Sub-theme 4: Basic daily care.**   In simple basic nursing, the work of nursing assistants is taken over by professionals with relevant medical background:

"*Our staff in Wuhan and the previous work content was different, the main task was disinfec-tion and sterilization, floor disinfection and sterilization. Floors, elevators, corridors, hand-rails, these disinfection work. The second one was to deliver meals to the elderly. After we picked up the three meals a day in the elevator, we would deliver them to the elderly's room, then one was the old people's drugs. For some old people with multiple chronic diseases, they need to dispense drugs" (Participant 1).*

## Main theme 4: Prevention and management strategies

**Sub-theme 1: Establish contingency plans.**   The participants showed formulating service specifications, standard processes and relevant manuals, so as to make everyone clear their responsibilities and work rules was essential:

"*I think the most outstanding work done by our team, focused on the development of epidemic prevention standards and work processes, through the work norms, risk aversion, and were picturesque, such as the removal of the protective clothing this step all picture and text attached to the wall, the corridor paste a string of walk, you walked over, you could complete the removal of this protective clothing (see the S1 Appendix)"* (Participant 1).

**Sub-theme 2: Health assessment.** Most of the elderly in institutions suffer from basic diseases such as chronic diseases, and may have many different symptoms. During the epidemic period, we should do a good job in supervision and comprehensive evaluation.

"*During the epidemic period, a grandmother with cerebral infarction always said that she was suffering from physical pain one night. At the time of shift handover, a nurse suddenly found out that it was wrong, and suggested that the doctor do a brain CT, which found that there was a thrombus in the cerebral blood vessels, and made corresponding treatment in time*" (Participant 1).

**Sub-theme 3: Targeted training.** The training targets are mainly divided into three categories, management, medical workers, and institutional logistics support (such as nursing assistant, doormen, cleaners, cooks, etc.):

"*We were basically put all the institutional processes on their walls, and they've got them all on the wall, and we had led them to talk and do it, and they were supposed to be able to learn to stick some of the more important points on the wall. All systems, all processes, that was, traditional protective processes, were all posted to them in a unified corridor, and they learn by themselves*" (Participant 5).

Training begins immediately after the team of experts has developed relevant processes and systems in the context of the organization:

"*In terms of personnel supervision, most of them were nursing assistants, because their overall knowledge of nursing assistants was weak and their knowledge of prevention and control was relatively lacking. So, we first did one thing, full staff training, including the relevant knowledge of COVID-19 and related knowledge of protection, spread the relevant knowledge*" (Participant 4).

**Sub-theme 4: Single isolation.** For confirmed cases immediately sent to designated hospitals for isolation treatment, their previous living environment is strictly sterilized and sterilized:

"*When I first found out that there was an old man who was diagnosed, all the old people were living in different layers at the first time. No old people lived in the same room. At the beginning, they were two old people living in the same room. Then they had a confirmed case at the first time and isolated all the old people like a single room*" (Participant 7).

**Sub-theme 5: Restrict visitor.** Strict entry and exit management system, entry and exit personnel, vehicles and materials must be strictly controlled:

"*For those who had to go in and out of the agency, vehicles carrying supplies were registered and temperature measured. The entry personnel block the face, by the special person used 500*

*mg/L chlorine disinfectant top-down Z shape to spray the whole body and the sole and carries on the hand disinfection to enter......" (Participant 2).*

### Main theme 5: Strategies for public health emergencies

**Sub-theme 1: Physical space design.** The nursing home must be equipped with three areas and two channels when they are established. The distance between floors should not be too close:

"*So when I talk about the old facilities of the nursing home, I mean that, when they design pension institutions, they do not better involve human concepts" (Participant 3).*

**Sub-theme 2: Personal allocation.** The staff of the organization should be in a proper proportion. Medical staff and nurses should form a small team to serve the elderly together.

"*The organization should have medical staff, only nursing assistants and managers are not enough, they do not have enough professional knowledge to face public health events such as epidemic" (Participant 2).*

**Sub-theme 3: Routine training.** Regular training and emergency plan drill shall be conducted at ordinary times to clarify emergency measures, prevention and control process:

"*These materials for killing may need to be stored. Then there are even special personnel to train the use and operation of these things, and there are training and supervision, which may have been used as a normalized epidemic outbreak season, and then as a normalized killing" (Participant 5).*

### Discussion

As far as we know, this is the first qualitative research on the experience of first-line medical staff who supported the nursing homes in Wuhan since the occurrence of COVID-19. Through the interview with medical staff in different positions back from Wuhan, we found the main difficulties faced by the nursing homes, the psychological experience of the medical staff and older people and the needs for improving the service for the older people.

Our study found that due to the sudden and rapid spread of this epidemic, the lack of materials and the weak knowledge of the original personnel, there is not only a large shortage of medical staff, but also the weakness of the ability in coping with disinfection and sterilization in the nursing homes. In addition, the nursing staff are lack of understanding of the epidemic situation and corresponding training, and the consciousness of disinfection, isolation and protection are relatively weak [18]. To prevent the risk of cross-infection, we encourage nursing homes to set up at least one isolation and observation room on each floor. Three areas and two passageways with an area of not less than 25 square meters between the living area and the nursing area, and the three areas are important [19]. Meanwhile, staff should complete the removal of protective clothing in these areas. The content of relevant knowledge made into pictures, pasted on the wall for learning at any time for both the staff and the elderly, previous study have shown that this method was more conducive to improving the sense of isolation of staff and clients [20].

The results of this study showed that in addition to the lack of knowledge in dealing with the COVID-19, the staff also showed psychological problems. Whenever a new infectious

disease with wide infectivity and high mortality appears, there would be some psychological pressure on the public [21, 22]. In view of this, first of all, reasonably allocate personnel and epidemic prevention teams including doctors, nurses, caregivers and managers should be set up. Secondly, timely and effectively training should be made, and COVID-19 knowledge education should be infiltrated into daily work. For example, a short minutes of every-day summary meeting can be held to increase the communication between staffs. Many studies showed that staffing characteristics and appropriate management, including staffing levels, turnover, professional staff mix, and use of every-day summary meeting, were proposed to influence quality of care on nursing homes [23–25]. Effective staffing and management can improve the ability of nursing homes to face public emergency.

We also found that there is a large shortage of medical supplies in many nursing homes. Nursing homes should establish and improve the emergency reserve system of medical materials, manage protective materials scientifically, and ensure that emergency supplies are adequate. The management department should always pay attention to the needs of front-line medical staff, reasonably arrange the distribution of protective materials, reduce the waste of protective equipment to ensure that front-line nursing staff have enough protective equipment.

While implementing nursing homes closure management, Taoyuan County of China has established a "four 'one'" working mechanism, which may be able to create an epidemic prevention "safety island" for centralized support for the older people. First, to take the body temperature once a day for the older people. Then, the knowledge of prevention and control be preached every day. Third, carry out disinfection once a day. And the last "one" is, video feedback on epidemic prevention will be carried out once a day. This could be a good example for the regulation of nursing homes.

Due to the sudden outbreak of the epidemic and the high mortality rate, the front-line medical staff had higher mental stress. A survey on the psychological problems of nurses showed that 85.37% of the front-line medical staff have certain psychological problems even they are honored to participate in the front-line work [26]. This is consistent with the results of our interview. Attention should be paid to the emotional experience, intervene and guide them in time. At the same time, paying attention to the adequacy of their rest time and the diet. Because the high sense of social responsibility, awe of life and professional values of saving lives and healing the wounded, timely encouragement and reward will contribute to their work [27–29].

We noticed that because the family members were unable to visit the nursing homes during the epidemic, the older people generally showed more psychological vulnerability. A considerable number of older people have varying degrees of paranoia and cognitive impairment, and even need drug control, but previous results show that the company of family members can ease the tension and uneasiness of the elderly, which required medical staff to be of great patience and spend a lot of energy to explain and appease them [30–32]. Some family members were not cooperative with the closure measures, all of which greatly increase the intensity of work. Our results show that the anxiety of the elderly can be reduced by contacting their families through video phone; putting green plants in the room and making greeting cards for the elderly also have a certain effect on relieving stress.

A limitation of this study was that the number of interviewees is small and there was situation that two interviewees went to the same nursing home. But in the methodological of view, the rigor of this study depends on reflexivity, rather than recruiting a larger number of participants. One of the advantages of this study is that it showed the ability to collect rich data from small target groups, embodies the value of qualitative research, and replaces the traditional face-to-face interview of qualitative research with a way of online video. Taking into account

of the advantages and limitations, participants included in this study were staff with certain medical knowledge background, future research can be carried out on nursing staff and managers of nursing homes who have no medical professional background knowledge to find ways in improving the professional knowledge and work efficiency that are suitable for nursing homes.

## Conclusion

The sudden onset of COVID-19 in the whole world led to an immediate necessity to study this phenomenon, as elder are the most susceptible population, nursing homes and health care professionals are in an urgent situation to adapt to this new epidemic environment. The results of the current study of the first batch of medical staff to Wuhan reflect the subjective experience of their interventions with older people and nursing homes, indicate that there are a lot we can do as medical staff to make the older people live better lives in nursing homes.

## Implications for nursing management

On condition that the reasonable environment regulations, organizational arrangements, and knowledge spreading about COVID-19 are not carried out, the prevention and control outcomes are extremely hard to meliorate. It is necessary to build and provide a safe working environment that the nursing staff can work in peace. Findings will inform managers of some reasonable instructional strategies for implementing effective infection management. Nursing homes need to provide targeted services to help alleviating their bad psychology for residents.

## Supporting information

**S1 Appendix. The date of this study.**
(DOCX)

## Author Contributions

**Data curation:** Xiu-Fen Yang, Meng-qi Li, Lu-lu Liao, Shuang Wu, Ping Yin.

**Formal analysis:** Meng-qi Li, Shuang Wu.

**Investigation:** Si Zhao.

**Methodology:** Lu-lu Liao.

**Software:** Xiu-Fen Yang.

**Writing – original draft:** Xiu-Fen Yang, Meng-qi Li, Lu-lu Liao.

**Writing – review & editing:** Hui Feng, Si Zhao, Ping Yin.

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
