## [Decision Letter · Decision Letter 0]

8 Jan 2021

PONE-D-20-36035

A qualitative study of the first batch of medical assistance team's first-hand experience in supporting the nursing homes in Wuhan against COVID-19

PLOS ONE

Dear Dr. FENG,

Thank you for submitting your manuscript to PLOS ONE. After careful consideration, we feel that it has merit but does not fully meet PLOS ONE’s publication criteria as it currently stands. Therefore, we invite you to submit a revised version of the manuscript that addresses the points raised during the review process.

We look forward to receiving your revised manuscript.

Kind regards,

Giovanni Ottoboni

Academic Editor

PLOS ONE

Journal Requirements:

2. Please include a copy of the interview guide used in the study, in both the original language and English, as Supporting Information, or include a citation if it has been published previously.

3. Thank you for including your ethics statement: 

"Human participants:Support staff of Wuhan elderly care institutions

Field research: Institution:School of Nursing, Central South University. Nursing and Behavioral Medicine Research Ethics Review Committee

number:E202037

Form of obtaining consent: Because we are conducting qualitative research, the data source is mainly qualitative interview. Before conducting the interview, we will explain our research purpose and data usage to the interviewee, and obtain the interviewee’s oral consent. The interview will be conducted later, otherwise the interview will not continue.".   

a. Please amend your current ethics statement to confirm that your named institutional review board or ethics committee specifically approved this study.

4. Please amend your current ethics statement to address the following concerns: Please explain why written consent was not obtained, how you recorded/documented participant consent, and if the ethics committees/IRBs approved this consent procedure.

5. Please include your tables as part of your main manuscript and remove the individual files. Please note that supplementary tables should be uploaded as separate "supporting information" files.

Additional Editor Comments:

Although the topic the authors attempt to report is both critical and timely, I agree with the reviewers' comments.

The paper needs to be reviewed drastically and proofread by a native English speaker: it is challenging to read as it is.

The authors must address their attention on the methods part, capitalize on the COREQ guidelines on the EQUATOR network, and implement the discussion according to the most recent literature on the argument.

Reviewers' comments:

Reviewer's Responses to Questions

**Comments to the Author**

1. Is the manuscript technically sound, and do the data support the conclusions?

Reviewer #1: No

Reviewer #2: Partly

Reviewer #3: Partly

2. Has the statistical analysis been performed appropriately and rigorously? 

Reviewer #1: N/A

Reviewer #2: N/A

Reviewer #3: N/A

3. Have the authors made all data underlying the findings in their manuscript fully available?

Reviewer #1: No

Reviewer #2: Yes

Reviewer #3: Yes

4. Is the manuscript presented in an intelligible fashion and written in standard English?

Reviewer #1: No

Reviewer #2: Yes

Reviewer #3: No

5. Review Comments to the Author

Reviewer #1: This manuscripts describes a study on the experiences of frontline care home staff during the COVID-19 pandemic. Reading the manuscript was difficult, and there were several sections missing and lacking academic rigour. I have numerous comments about the manuscript, which are listed below:

- The abstract is full of grammatical mistakes and typos, and also does not contain a Results section. The identified themes should not be listed in the Methods section. The sample size is too small to warrant publication in the form of a paper - instead this may be more suitable, given strong re-edit of language, in the form of a letter.

- Equally, the Introduction is very difficult to read due to language issues, and suitable references are missing. The wording is not appropriate for academic standards. After this early impression, I have stopped reading as both the abstract and introduction indicate that the paper is unfortunately not suitable for publication.

Reviewer #2: The paper is very interesting but the sample is small; no tools are highlighted to detect the psychological problems of residents and staff. In teh interviews the authors could insert a references of Standardized tools to detect burn out or burden.

Reviewer #3: Thank you for giving me the opportunity to review your manuscript. This is a very topical paper, addressing a very important area as we are in the grasp of the global COVID pandemic. In general, I think that the standard of English (grammar & spelling) could be greatly improved with proof reading/editing by a native English speaker. At present, the level of English is not of an appropriate standard. I think there is scope for a higher level of rigour in the reporting in this manuscript/study. The author(s) may benefit from giving some attention to the COREQ guidelines on the EQUATOR network.

I think some more work is required on the discussion component/section of this manuscript, the author(s) do not really critique to relevance of their findings in relation to previous literature. Finally, it is not clear whether the study examines medical staff or nursing staff or even untrained staff. More clarity should be given to what is meant by 'medical assistance team'.

6. PLOS authors have the option to publish the peer review history of their article (what does this mean?). If published, this will include your full peer review and any attached files.

Reviewer #1: No

Reviewer #2: No

Reviewer #3: **Yes: **Professor Graeme D. Smith

---

## [Author Response · Author response to Decision Letter 0]

3 Mar 2021

We have uploaded a file separately as response to reviewers. You can see my reply in that file more clearly.

Rebuttal letter

Title of article: A qualitative study of the first batch of medical assistance team's first-hand experience in supporting the nursing homes in Wuhan against COVID-19 (PONE-D-20-36035R1)

 We would like to thank the reviewers of their comments. There are many changes in overall structure and the language, in order not to cause visual confusion, we have only marked the revisions in red in the revised manuscript for comments of reviewer and editor. Our responses to the issues raised are listed below:

Editor’s Review comments Author response to comments

 We have made appropriate changes to the whole article. Thanks.

2.Please include a copy of the interview guide used in the study, in both the original language and English, as Supporting Information, or include a citation if it has been published previously We have modified it, include a citation has been published previously (see page 8 line 7).

3. Thank you for including your ethics statement:

Please amend your current ethics statement to confirm that your named institutional review board or ethics committee specifically approved this study. Please amend your current ethics statement to address the following concerns: Please explain why written consent was not obtained, how you recorded/documented participant consent, and if the ethics committees/IRBs approved this consent procedure. Because the project was interviewed in the most serious time of COVID-19. The whole Chinese people were isolated at home and were not allowed to gather. So we adopted video interviews, and did not directly contact with the interviewees.However, before conducting the interview, we explained the research purpose and data usage to the interviewee, and the oral consent were all obtained. Ethical Approval was gained from the Nursing and Behavioral Medicine Institutional Review Boards, Xiangya School of Nursing, Central South University (Approval number: E202037).

4.Please include your tables as part of your main manuscript and remove the individual files. Please note that supplementary tables should be uploaded as separate "supporting information" files. We have modified it. Thanks.

5.The paper needs to be reviewed drastically and proofread by a native English speaker: it is challenging to read as it is. After making the changes, we asked an American nursing professor from the University of California to make the appropriate language changes. She is a Native American. Thanks!

6.The authors must address their attention on the methods part, capitalize on the COREQ guidelines on the EQUATOR network, and implement the discussion according to the most recent literature on the argument.

 We have made a lot of changes to the method part according the the COREQ guidelines on the EQUATOR network (see page 6-10). Thanks!

Reviewers' comments

Reviewers' comments

 Author response to comments

1. Is the manuscript technically sound, and do the data support the conclusions?

 We make a new analysis of the interview data and draw more sufficient evidence to support our conclusion.Thanks! 

2. Has the statistical analysis been performed appropriately and rigorously?

 We have revised the legal part of the other party. Before, we used the theme analysis method. After our group's new discussion, we adopted the content analysis method to make a new analysis of the interview results. You can see the analysis of method (see page 9-10).

3. Have the authors made all data underlying the findings in their manuscript fully available?

 We have added more supporting materials to support our research. Thanks!.

4. Is the manuscript presented in an intelligible fashion and written in standard English?

 After making the changes, we asked an American nursing professor from the University of California to make the appropriate language changes. She is a Native American. Thanks!

Reviewer #1

Reviewer #1’s comments Author response to comments

- The abstract is full of grammatical mistakes and typos, and also does not contain a Results section. 

 We have modified it (see page 2-3). Thanks! 

- The identified themes should not be listed in the Methods section. We put it in the results section and attached the table 2. Thanks! 

- The sample size is too small to warrant publication in the form of a paper - instead this may be more suitable, given strong re-edit of language, in the form of a letter. Although the sample size is small, we did not terminate the interview until no new Main theme appeared. One of the advantages of this study is that it showed the ability to collect rich data from small target groups, embodies the value of qualitative research, and replaces the traditional face-to-face interview of qualitative research with a way of online video. Taking into account of the advantages and limitations, participants included in this study were staff with certain medical knowledge background, future research can be carried out on nursing staff and managers of nursing homes who have no medical professional background knowledge to find ways in improving the professional knowledge and work efficiency that are suitable for nursing homes. In addition, the interviewers also provided us with their training materials, emergency plans and management systems during the epidemic period for our reference. Thanks!

- Equally, the Introduction is very difficult to read due to language issues, and suitable references are missing. The wording is not appropriate for academic standards. After this early impression, I have stopped reading as both the abstract and introduction indicate that the paper is unfortunately not suitable for publication.

 I'm very sorry to have caused so much trouble. After carefully reading your comments on this part, we have made appropriate amendments. Thanks!

Reviewer #2

Reviewer #2’s comments Author response to comments

- The paper is very interesting but the sample is small; 

 Although the sample size is small, we did not terminate the interview until no new Main theme appeared. One of the advantages of this study is that it showed the ability to collect rich data from small target groups, embodies the value of qualitative research, and replaces the traditional face-to-face interview of qualitative research with a way of online video. Taking into account of the advantages and limitations, participants included in this study were staff with certain medical knowledge background, future research can be carried out on nursing staff and managers of nursing homes who have no medical professional background knowledge to find ways in improving the professional knowledge and work efficiency that are suitable for nursing homes. In addition, the interviewers also provided us with their training materials, emergency plans and management systems during the epidemic period for our reference. Thanks!

- No tools are highlighted to detect the psychological problems of residents and staff. In teh interviews the authors could insert a references of Standardized tools to detect burn out or burden.

 According to the information from the interview materials, the Chinese version SAS was as a standardized tools to detect burn out or burn in nursing homes(See page 13: line 8-16).

Reviewer #3

Reviewer #3’s comments Author response to comments

- In general, I think that the standard of English (grammar & spelling) could be greatly improved with proof reading/editing by a native English speaker. At present, the level of English is not of an appropriate standard. I think there is scope for a higher level of rigour in the reporting in this manuscript/study. 

The author(s) may benefit from giving some attention to the COREQ guidelines on the EQUATOR network.

 We have made a lot of changes to the method part according the the COREQ guidelines on the EQUATOR network. After making the changes, we asked an American nursing professor from the University of California to make the appropriate language changes. She is a native English speaker. Thanks!

- I think some more work is required on the discussion component/section of this manuscript, the author(s) do not really critique to relevance of their findings in relation to previous literature. 

 We modified the discussion part appropriately and added some new literatures In order to support our conclusion, Thanks!

Finally, it is not clear whether the study examines medical staff or nursing staff or even untrained staff. More clarity should be given to what is meant by 'medical assistance team'.

 In our study , purposive sampling was used to select staffs from first line medical assistance team with different occupations, who supported the nursing homes in Wuhan against COVID-19 at the time of the most serious epidemic and lack of manpower , and to ensure a broad perspective of medical members of different positions. The first-line medical assistance team consisted of 26 members, including clinical nurses, doctors, nursing managers and nursing assistants. The medical assistance team were mainly from the department of infection, respiratory and critical care medicine, cardiovascular department, neurology department and other departments, forming a general practice team, which undertake the treatment and management of a nursing home, reducing contact with the outside. They all have many years of clinical experience, before they enter the nursing home, they will be specially trained by the medical staff of the infection department. (see page 7: line 2-12 ).

---

## [Decision Letter · Decision Letter 1]

23 Mar 2021

A qualitative study of the first batch of medical assistance team's first-hand experience in supporting the nursing homes in Wuhan against COVID-19

PONE-D-20-36035R1

Dear Dr. FENG,

We’re pleased to inform you that your manuscript has been judged scientifically suitable for publication and will be formally accepted for publication once it meets all outstanding technical requirements.

Kind regards,

Giovanni Ottoboni

Academic Editor

PLOS ONE

Additional Editor Comments (optional):

Reviewers' comments:

Reviewer's Responses to Questions

**Comments to the Author**

1. If the authors have adequately addressed your comments raised in a previous round of review and you feel that this manuscript is now acceptable for publication, you may indicate that here to bypass the “Comments to the Author” section, enter your conflict of interest statement in the “Confidential to Editor” section, and submit your "Accept" recommendation.

Reviewer #2: All comments have been addressed

Reviewer #3: All comments have been addressed

2. Is the manuscript technically sound, and do the data support the conclusions?

Reviewer #2: Yes

Reviewer #3: Yes

3. Has the statistical analysis been performed appropriately and rigorously? 

Reviewer #2: Yes

Reviewer #3: (No Response)

4. Have the authors made all data underlying the findings in their manuscript fully available?

Reviewer #2: Yes

Reviewer #3: No

5. Is the manuscript presented in an intelligible fashion and written in standard English?

Reviewer #2: Yes

Reviewer #3: Yes

6. Review Comments to the Author

Reviewer #2: Corrections made from the previous manuscript are good and acceptable for publication.The methodology used was explained and the writing style in English was improved. In my opinion, the procedure for assessing the conditions of the elderly should be better detailed through the comprehensive geriatric assessment (CGA) but ( being a qualitative study ) the absence of this methodology can be accepted for the objectives of this study.

Reviewer #3: Thank you for addressing most of the recommendations within my original review. I do wonder if you could have given greater emphasis to the role that nursing staff played within your study.

7. PLOS authors have the option to publish the peer review history of their article (what does this mean?). If published, this will include your full peer review and any attached files.

Reviewer #2: No

Reviewer #3: **Yes: **Professor Graeme D. Smith

---

## [Editor Report · Acceptance letter]

29 Mar 2021

PONE-D-20-36035R1 

A qualitative study of the first batch of medical assistance team's first-hand experience in supporting the nursing homes in Wuhan against COVID-19 

Dear Dr. Feng:

I'm pleased to inform you that your manuscript has been deemed suitable for publication in PLOS ONE. Congratulations! Your manuscript is now with our production department. 

Kind regards, 

on behalf of

Dr. Giovanni Ottoboni 

Academic Editor

PLOS ONE